# Novel Fuzzy Logic Scheme for Push-Based Critical Data Broadcast Mitigation in VNDN

**DOI:** 10.3390/s22208078

**Published:** 2022-10-21

**Authors:** Sajjad Ahmad Khan, Huhnkuk Lim

**Affiliations:** Department of Computer Engineering, Hoseo University, Asan-si 31499, Korea

**Keywords:** push-based forwarding, critical data packet, vehicular named data networking (VNDN), K-means clustering, fuzzy logic, cluster head (CH)

## Abstract

Vehicular Named Data Networking (VNDN) is one of the potential and future networking architectures that allow Connected and Autonomous Vehicles (CAV) to exchange data by simply disseminating the content over the network. VNDN only supports a pull-based data forwarding model, where the content information is forwarded upon request. However, in critical situations, it is essential to design a push-based data forwarding model in order to broadcast the critical data packets without any requests. One of the challenges of push-based data forwarding in VNDN is the broadcasting effect, which occurs when every vehicle broadcasts critical information over the network. For instance, in emergency situations such as accidents, road hazards, and bad weather conditions, the producer generates a critical data packet and broadcasts it to all the nearby vehicles. Subsequently, all vehicles broadcast the same critical data packet to each other, which leads to a broadcast storm on the network. Therefore, this paper proposes a Fuzzy Logic-based Push Data Forwarding (FLPDF) scheme to mitigate the broadcast storm effect. The novelty of this paper is the suggestion and application of a fuzzy logic approach to mitigate the critical data broadcast storm effect in VNDN. In the proposed scheme, vehicles are grouped into clusters using the K-means clustering algorithm, and then Cluster Heads (CHs) are selected using a fuzzy logic approach. A CH is uniquely responsible for broadcasting the critical data packets to all other vehicles in a cluster. A Gateway (GW) has the role of forwarding the critical data packets to the nearest clusters via their GWs. The simulation results show that the proposed scheme outperforms the naive method in terms of transmitted data packets and efficiency. The proposed scheme generates five times fewer data packets and achieves six times higher efficiency than the naive scheme.

## 1. Introduction

Due to the advancement in Information and Communication Technology (ICT), Autonomous Vehicles (AVs) enter the commercial market and progress toward full autonomy. Connected and Autonomous Vehicles (CAVs) combine the features of AVs and Connected Vehicles (CVs), providing more benefits than their individual functionalities. CAVs are equipped with smart cameras, sensors, control systems, and automated applications. Some practical applications of CVs include traffic and congestion control, fleet management, and onboard device-assisted multimedia systems [1]. In order to support the Advanced Driver Assistance Systems (ADAS), the automobile industry has focused on the research and development of CAVs, particularly Vehicle-to-Everything (V2X) communication. Therefore, most developed countries continue to improve their Intelligent Transportation Systems (ITS) in order to manage people, autonomous vehicles, and roads using cutting-edge technologies [2]. ITS enable road monitoring, traffic control, vehicular communication, smart decisions during emergency situations, etc. With regard to the future of ITS, traffic management and road safety applications play an essential role in the control of traffic congestion and the reduction of accidents [3].

The Vehicular Ad Hoc Network (VANET) is regarded as one of the fundamental architectures of ITS since it provides valuable services. A VANET is a collection of moving or stationary vehicles connected via a wireless network. They manage intelligent traffic and provide real-time information as well as event collections. Initially, the main objective of VANET was to speed up the development and use of self-driving cars by providing reliable communication with the help of ITS. Nevertheless, it has evolved into a special kind of networking architecture that allows CAVs to communicate with each other and the road infrastructure. In VANET, CAVs are responsible for sending, receiving, and relaying information through V2X communication channels [4]. VANET delivers information and entertainment services known as infotainment, which requires ultra-reliable and low-latency communication models. VANET is adopted to improve road safety, control traffic congestion, regulate data flow, and facilitate emergency services through the reliable and timely dissemination of alerts and warnings. It also allows CAVs to check the weather and play music and videos on the way [5].

VANET inherits the features of the Mobile Ad Hoc Network (MANET) and is based on Telecommunication Control Protocol/Internet Protocol TCP/IP standards [6]. TCP/IP is a widely adopted communication standard for wired and wireless communications. For instance, the technologies used in the Internet of Things (IoT) and Internet of Vehicles (IoV) depend heavily on the underlying TCP/IP standards [7]. Multiple radio access methods, including Wireless in Vehicular Environments (WAVE), Dedicated Short Range Communications (DSRC), Long-Term Evolution Vehicle-to-Everything (LTE-V2X), and Cellular Vehicle-to-Everything (C-V2X), all operate under the TCP/IP standards [8]. However, the TCP/IP paradigm in VANET for the distribution of large amounts of data is challenging. In other words, TCP/IP-based communication cannot meet the requirements of a VANET due to the vehicles’ frequent mobility and the dense environments [9].

Furthermore, security warnings and beacon messages that need timely communication necessitate a latency-free system. Moreover, quick content delivery, vehicle mobility management, a robust traffic management system, and adaptive routing render the TCP/IP infrastructure infeasible and make it complicated to meet the quality of service needs of various applications. Other obstacles include unstable communication between vehicles as a result of continuous changes in the networks as well as network instability. It is challenging to maintain dedicated pathways between source and destination vehicles in a TCP/IP-based VANET due to the mobility of vehicles [10].

On the other hand, a Named Data Network (NDN) is designed under the umbrella of Information-Centric Networking (ICN) [11]. NDN is a content-oriented architecture that has the potential to replace IP-based architecture in the near future. In the NDN, the data can be accessed according to their name/content/prefix rather than their IP address (host location) [12,13]. The content’s name is a key element in NDN-based communication in contrast to the IP address. Due to the benefits of NDN over traditional IP-based networking, new designs have been created to implement the VANET by using the NDN architecture [14]. Named Data Network (NDN)-enabled VANET is known as Vehicular NDN (VNDN), which has emerged as a viable data distribution option for connected vehicles (cars) [15]. Instead of safeguarding a communication connection, NDN encrypts data. However, it lacks defenses against Denial of Service (DoS) and Distributed-DoS (DDoS) attacks [16]. The flooding effect of interest packets is one of the DDoS attacks in VNDN in which the attacker sends many bogus requests and prevents total submissions from legitimate vehicles. This study is motivated by a desire to mitigate the data packet broadcast storms of unregulated and unwanted traffic over VNDN.

In VNDN, vehicles may be consumers, producers, or intermediate vehicles. By design, VNDN supports only a pull-based communication model in which a consumer vehicle creates an interest packet and broadcasts it on the network [17]. The intermediate vehicles forward the interest packet to other vehicles in the network until it reaches the producer. After receiving the interest packet, the vehicle (producer) of the content or an intermediate vehicle that has stored the requested content in its cache responds with the data packet. The data packet of the requested content is sent to the consumer vehicle simply by broadcasting it to all the nearby vehicles [18]. In order to realize critical situations such as accidents, bad weather conditions, and traffic jams, it is essential for the critical data packets to be transmitted in a push-based manner [19,20]. In push-based data forwarding scenarios, a vehicle (producer) generates a critical data packet and transmits it to all the vehicles (consumers) without their requests. A push-based data forwarding scheme is thus required to be designed in VNDN. To the best of our knowledge, some researchers have offered content retrieval solutions based on a push-based strategy for naive NDN, but not for VNDN [21]. For instance, the authors of [22] suggested a distributed push-based caching management scheme called Push-Based Traffic-Aware Distributed Cache (P-TAC) in NDN. When interest is received, a Content Store (CS) in P-TAC requests one of its neighbors to store the content on its behalf.

Similarly, the authors of [23] provided a system to enable push-based Internet of Things (IoT) traffic based on three situations. The authors adopted three distinct scenarios in order to manage various sorts of IoT traffic, such as interest notification, unsolicited data, and virtual interest polling. In [24], the authors created a system for secure sensing using NDN-enabled sensors. The authors regarded an environmental sensing device as a producer. When new material becomes accessible, the sensor alerts the consumer’s application for it to broadcast an interest packet and retrieve the content. All of these methods need customer interest before they can start exchanging content, which creates too much latency in NDN-based communication.

Concerning the current forwarding strategies that enable push-based critical data forwarding in the NDN environment, the critical data packets are broadcast and transferred without proper procedures. As a result, critical data dissemination creates a broadcast storm effect in push-based NDN communication. In terms of mitigating the critical data broadcast storm effect, the research is at its initial phase, and comprehensive research is required on this particular topic. The only research that exists on the mitigation of the critical data broadcast storm effect is presented in [25], which was applied in IoT-based NDN communication. Therefore, we propose a novel scheme based on a fuzzy logic approach to mitigate the critical data broadcast storm. The proposed scheme is the first research work of its kind. We designed a fuzzy logic approach, applied it in VNDN, and reduced the broadcast storm effect resulting from the broadcasting of the critical data packets. In this regard, we assumed critical situations, and the critical data packets were transmitted through a push-based mode of communication. Our proposed model can be used as a starting point for future push-based communication in which the critical data packets are disseminated in the VNDN environment.

The rest of the paper is structured as follows. Section 2 explains the Vehicular Named Data Networking (VNDN) architecture to understand the context. The push-based data forwarding procedure used in VNDN is illustrated in Section 3. Section 4 demonstrates the proposed push-based data forwarding scheme with fuzzy logic. The simulation environment and results are discussed in Section 5. Finally, Section 6 concludes the paper.

## 2. Vehicular Named Data Networking (VNDN)

The Information-Centric Network (ICN) project called Named Data Networking (NDN) is a new architecture designed to upgrade traditional TCP and IP networks to content-based networks. The primary purpose of the modification of the network is to make it easier for data packets in a communication network to refer to their content names instead of their destination IP addresses [26]. Recently, most research has focused on integrating vehicular communication with the NDN architecture, known as Vehicular Named Data Networking (VNDN). In other words, VNDN is an application class of VANETs and NDN in which a group of vehicles is connected through the On-Board Units (OBUs) and obtains access to a centralized data network. An OBU allows vehicles to process, store, and communicate with each other. In addition, OBUs can be connected wirelessly with or without Road Side Units (RSUs), enabling them to form ad hoc, dispersed, and self-organized networks. The major objectives of VANET using the NDN architecture are the rigorous exchange of information and the effort to increase road safety while improving the driving experience. In VNDN, vehicles can communicate and exchange interest and data packets via V2V and V2RSUs [27]. If a consumer vehicle needs any content, it generates a request (interest) with the name of the content and broadcasts it on the network. The intermediate vehicles rebroadcast the interest until it reaches the data producer. Once the interest packet reaches a node/vehicle that possesses the requested data, i.e., the name of the interest is the same as the name of the data or a prefix of the name of the data, the data are sent to the requested consumer vehicle. Since there are no source and destination addresses in a VNDN packet, the intermediate vehicles use the name and prefix in the Content Store (CS), Pending Interest Table (PIT), and Forwarding Information Base (FIB) to exchange interest and data packets [28]. Figure 1 demonstrates the general method used in the VNDN-based architecture, where the third layer makes use of a content name instead of an IP address.

### 2.1. Content Store (CS)

A Content Store (CS) is one of the important data structures used in VNDN-based communication. CS temporarily stores the data packets when it receives from the producers or intermediate vehicles. In VNDN-based communication, data packets are meaningful and need to be stored in a cache, i.e., CS, which can be forwarded to the consumer upon request. In order to maintain the updated version, the most recently used data packet can replace the old one. The vehicle checks the data packet in its CS when an interest packet is received.

### 2.2. Pending Interest Table (PIT)

Pending Interest Table (PIT) is another data structure of the VNDN architecture. PIT contains all the requested interest packets received and forwarded in the search for a data packet. Each PIT records the name or prefix of the data carried by the interest packet of its incoming and outgoing interface(s). After successfully delivering the data packet, the entry is removed from the PIT table.

### 2.3. Forwarding Information Base (FIB)

Forwarding Information Base (FIB) is a data structure used in the VNDN architecture. FIB generates a routing table that links content names to interfaces. A name-prefix-based routing protocol populates the FIB, which might have many output interfaces for each prefix.

### 2.4. Consumer Vehicle

In VNDN, any node/vehicle that generates an interest packet (request) for any content or information is called a consumer vehicle. Based on this NDN forwarding strategy, a consumer initiates an interest packet and broadcasts it to all the neighboring vehicles. The data packet will be received if the interest is fulfilled by the intermediate vehicles or the producer. The consumer vehicle is hence the data requester in a VNDN system.

### 2.5. Producer Vehicle

The node or vehicle that contains the original data in its depository is the producer in the VNDN. When an interest packet (request) is received at the producer, a data packet is prepared after matching the name or prefix of the content. The data are then forwarded to the consumer through the same path that the interest packet has traversed.

## 3. Push-Based Data Forwarding in VNDN

The VNDN basically supports a pull-based architecture in which communication is initiated by a consumer node/vehicle. However, in vehicular communication, there might be some critical situations in which push-based communication should be applied. These critical situations may include the broadcast of traffic information, advertisements, and critical alerts, among others. Figure 2 demonstrates a comparison of the pull-based and push-based data forwarding mechanisms in VNDN.

A VNDN is represented by the undirected graph $G_{t}\left(V,\in_{t}\right)$, where *V* is the collection of vertices and $\in_{t}$ indicates the links that connect the vehicles at time *t*. *V* denotes the connected vehicles in the network, such as $v_{1}$, $v_{2}$, $v_{3}$, …, $v_{N}$. *p* is the producer that contains the original content, and *c* is the consumer that intends to obtain that content. The symbol *O* refers to an object of the content that has two different kinds of chunks: the non-critical chunks consist of $n_{1}$, $n_{2}$, $n_{3}$, …, $n_{K}$, while the critical chunks include $r_{1}$, $r_{2}$, $r_{3}$, …, $r_{M}$. Using Equation (Equation 1), we can calculate the total transfer time for the data packet in a pull-based VNDN system [19]. We assume that the size of every chunk, regardless of the class, is the same and is denoted by $s_{r}$.
(1)$$\begin{array}{cc} T_{TCT}\left(G_{t\alpha },G_{tI},G_{tr},p,c\right)= & Q\left(G_{t\alpha },p,c\right)t_{\alpha }+Q\left(G_{tI},c,p\right)\left[\sum_{k=1}^{M}\left(t_{y}+\frac{S_{I}}{\tau_{y}}\right)\right]\\ \; & +Q\left(G_{tr},p,c\right)\left[\sum_{l=1}^{N}\left(t_{x}+\frac{S_{r}}{\tau_{x}}\right)\right] \end{array}$$
where $t_{x}$ and $\tau_{x}$ represent the latency and transfer rate for data chunks, whereas $t_{y}$ and $\tau_{y}$ represent the latency and transfer rate for interest packets. $Q(n_{1},n_{2})$ is the distance between two nodes (vehicles). $G_{t}\alpha$ represents the status of the network at the moment when a producer broadcasts the data content. $G_{tI}$ defines the condition of the network at the moment when an interest packet is broadcast. $G_{tr}$ represents the network status at the moment when a chunk of the data content is disseminated. $S_{I}$ and $S_{r}$ represent the length of interest and data packets, respectively. $t_{\alpha }$ indicates the time required for a node/vehicle to forward the data packet to its nearby nodes/vehicles.

On the other hand, Equation (Equation 2) can be used in the push-based communication model to calculate the total transfer time for critical data packets [19].
(2)$$T_{STT}\left(G_{t\beta },G_{tr},p,c\right)=Q\left(G_{t\beta },p,c\right)t_{\beta }+Q\left(G_{tr},p,c\right)\left[\sum_{l=1}^{N}\left(t_{x}+\frac{S_{r}}{\tau_{x}}\right)\right]$$
where, $t_{x}$ and $\tau_{x}$ represent the latency and transfer rate for the chunks of the critical data content. $Q(n_{1},n_{2})$ indicates the distance between two nodes (vehicles). $G_{t\beta }$ denotes the status of the network at the moment when a producer broadcasts the critical data content. $G_{tr}$ is the network status at the time when a producer transmits a chunk of critical data content. $S_{r}$ indicates the size of the critical data content. tβ is the time required for a node/vehicle to forward the critical data packet to its nearby nodes/vehicles.

## 4. Proposed Push-Based Data Forwarding Scheme with Fuzzy Logic

In order to mitigate the critical data packet broadcast storm, we propose a push-based data forwarding scheme using fuzzy logic that comprises several steps.

### 4.1. K-Means Clustering

The term “clustering” can be utilized to manage frequent communication between vehicles. In order to arrange the vehicles into groups, a K-means clustering approach is employed in this paper. K-means is an unsupervised machine learning approach that can be used for clustering. It is a technique that automatically divides vehicles into clusters or groups. Every vehicle is automatically assigned to a cluster upon receiving its Received Signal Strength Indicator (RSSI) value. However, each vehicle can only belong to one cluster. The center point of that cluster is called a centroid [29]. At first, the algorithm takes the number of clusters “K”. Then, “K” points are randomly selected as centroids. To find the optimum number for “K”, an Elbow algorithm is used.

Based on the centroids, the vehicles are grouped together. In the next step, central points are chosen to indicate the new centroids. The vehicles then group themselves according to the new centroids, and this process continues until no change in the group occurs. Figure 3 demonstrates the clustering with the use of the K-means clustering algorithm. The K-means algorithm follows the expected maximization to address the problem. After every second, these steps are repeated, allocating the vehicles’ locations to the nearest cluster. Each step calculates a new centroid for each cluster. The normalized function of “*J*” is calculated by using Equation (Equation 3).
(3)$$J=\sum_{i=1}^{m}\sum_{k=1}^{K}\omega_{ik}{∥|x_{i}-\mu_{i}∥|}^{2}$$
where $\omega_{ik}=1$ for data $x_{i}$ if it belongs to cluster *K*; otherwise, $\omega_{ik}=0$. Furthermore, $\mu_{k}$ is the centroid of each $x_{i}$ cluster. On the other hand, the normalized distances from centroids are calculated according to Equation (Equation 4).
(4)$$dist=\sqrt{{\left(x_{1}-y_{1}\right)}^{2}+{\left(x_{2}-y_{2}\right)}^{2}}$$
where *x* and *y* are the coordinates of each vehicle.

### 4.2. Selection of Cluster Head (CH) using Fuzzy Logic

Fuzzy logic is a strategy that utilizes more than one parameter to determine and select one node/vehicle to be the Cluster Head (CH). We maintain the membership functions in our proposed scheme, and fuzzy rules are applied during the selection process. During the fuzzification process, the fuzzy system converts the input values into fuzzy values. Input parameters are explained in the following subsections. The inference engine generates a fuzzy output according to the rules by entering the input parameters. During defuzzification, the calculated output value is converted into the actual value [30]. As a result, the CH is then selected based on the output value. The fuzzy logic procedure for selecting a CH is explained in Figure 4.

The following input parameters are used to choose the CH in a given cluster.

#### 4.2.1. Link Weight (Centrality)

In selecting the CH in a cluster, the link weight is an important parameter to be considered. During this step, the distances of all nodes/vehicles from the center point is calculated by employing the normalized distance Equation (Equation 5).
(5)$$dist_{vi}=\left[norm\left(\sqrt{{\left(x_{2}-x_{1}\right)}^{2}+{\left(y_{2}-y_{1}\right)}^{2}}\right)\right]\times 100$$

The shortest distance from the center position is given a higher weightage and is to be chosen as a CH. However, other parameters also impact the selection process.

#### 4.2.2. Speed (Mobility)

The speed of a node/vehicle is also an important parameter for selecting the CH. If the speed is higher, it will move quickly and change its position frequently. Therefore, a node/vehicle’s speed impacts the control system, especially the fuzzy logic controller. The speed and mobility of a node/vehicle can be calculated by Equation (Equation 6).
(6)$$speed_{vi}\left(m/s\right)=\left(\frac{speed_{vi}}{\max_{i=1}^{n}speed}\right)\times 100$$

Each vehicle’s speed vi is normalized by dividing it by the maximum speed. Subsequently, the percentage of all the speed is forwarded to the fuzzy logic controller.

#### 4.2.3. Direction (Towards Producer)

In this step, the direction of the node/vehicle is obtained by taking the slope. If the slope is zero degree, the node/vehicle is assumed to be moving toward the producer of the critical data packet. If the slope is 180°, the node/vehicle is believed to be moving in the opposite direction. In this case, the node/vehicle will not be entertained for the selection of the CH due to its direction. Equation (Equation 7) is used for the vehicle’s direction.
(7)$$slope_{vi}=\left\{\begin{array}{cc} 1, & \mathrm{if}\;slope_{vi}=0\\ 0, & \mathrm{if}\;slope_{vi}=180 \end{array}\right.$$

#### 4.2.4. Total Weight

After calculating the input values of a node/vehicle, we obtain the total weight for each node/vehicle in the cluster by using the following Equation (Equation 8):(8)$$Total_{vi}=\left(dist_{vi}+speed_{vi}\right)\times slope_{vi}$$
(9)$$CH=\min_{i}^{n}\left(Total_{vi}\right),\;\;\;\;\;CH\neq 0.$$

Any node/vehicle carrying the minimum value will be selected as the CH of that cluster. It is noteworthy that the value must be greater than zero in order to participate in the selection process.

### 4.3. Proposed Data Packet Format

In our push-based data forwarding scheme, we modified the existing data packet format and inserted some extra information to disseminate the critical content. The following extra fields can be inserted to modify/update the data packet in order to control the data broadcasting in the VNDN environment. The important fields of the proposed data packet are presented in Table 1.

#### 4.3.1. Alert Type

This is an important field of the data packet that can be used to understand the nature of the data packet. In critical scenarios, the alert will be highlighted as critical. Therefore, understanding the nature of the alert, the consumer vehicles will forward it without having an interest request in their PIT tables.

#### 4.3.2. Location of the Producer

This field carries the location of the producer when the data packet is generated. Using the location helps to find the direction of the critical data packet. The consumer vehicles will be able to take action according to the critical data packet using the location of the producer. In other words, if the alert comes in the direction of the vehicle, then prior action will be taken; otherwise, it will ignore and drop the alert.

#### 4.3.3. Timer

The timer also plays a vital role during the dissemination of the data packet in push-based data forwarding. The timer is used to forward the data packet for a specific amount of time. If the timer reaches the maximum time, the data packet will be dropped and forwarding will be stopped by the intermediate vehicles.

### 4.4. Proposed Scheme for Producer

In a critical situation, the push-based data packet is generated by the producer and forwarded to the CH by using Algorithm 1. Every vehicle advertises its location, speed, and direction to all its neighboring vehicles. During the fuzzy logic process, every node/vehicle in the cluster identifies the CH and cluster GWs. Therefore, we assume that the producer also belongs to a cluster and has established a connection with the CH. When the producer receives a regular interest packet for non-critical data content, it generates an ordinary data packet and forwards it to the requested vehicle. However, if there is a critical situation, the producer generates a critical data packet and forwards it to the CH of that cluster. The CH is solely responsible for broadcasting it to all the member vehicles in that cluster. The pseudo-code for the CH and GW vehicles is presented in Algorithm 2.
**Algorithm 1** The proposed push-based critical data forwarding algorithm at a producer.1:Possible events: (Startup, interest *i*, consumer *c*, critical content *b*)2:Case Event:3:**if** Producer == New **then**4:     advertise each non-critical content Object *O*5:     **if** non-critical content *i* **then**6:          send *i* to consumer *c*7:          **if** critical content *r* **then**8:               construct a critical data packet *b*9:               send critical content *b* towards CH10:         **end if**11:    **end if**12:**end if**

**Algorithm 2** The proposed push-based critical data forwarding algorithm at a consumer.
 Received Data Packet: (ID, Content, Alert Type, Location, Timer, Signature)2:**if** Alert Type == $b_{i}$ (critical data packet) **then**      read *b*4:      **if** Vehicle ($v_{i}$)== CH (Cluster Head)**then**            Broadcast bi6:            **if** Vehicle ($v_{i}$)== $GW_{i}$ **then**                  Forward *b* to $GW_{i+1}$8:            **else**                  drop *b*10:           **end if**      **end if**12:
**end if**



### 4.5. Proposed Scheme for Consumer

In push-based data forwarding, we assume that all the vehicles are consumer vehicles, and that the critical data content is crucial for them. Therefore, the critical data packet is forwarded to the CH of that cluster without receiving any request or interest packet. Every member vehicle of the cluster contains a MAC Table and cluster information such as CH and Gateways (GW). Upon receiving the critical data packet, the CH broadcasts the critical data packet to all the member vehicles in that cluster. If the intermediate vehicle is a GW, it forwards it to the GW vehicle of its neighboring cluster. If the consumer vehicle is only a member vehicle of that cluster, then it drops the data packet and does not forward it to any vehicle in its vicinity. Algorithm 2 explains the data forwarding procedures followed by the CH and intermediate vehicles as well as their roles during critical conditions.

### 4.6. Critical Data Forwarding Procedure in the Proposed Scheme

Figure 5 demonstrates the critical data flow from the producer to all other vehicles. We assume that the critical data packet, such as an emergency alert, is necessary for all vehicles moving towards the location of the critical situation. Before any critical condition, all the vehicles are grouped into different clusters by employing the K-means clusters, as discussed in Section 4.1. Subsequently, different roles are assigned to the vehicles in each cluster by using the fuzzy logic method, presented in Section 4.2.

When a critical situation occurs, the producer generates a critical data packet and transmits it to the CH using Algorithm 1. The producer sends the alert type, location, and timer to the critical data packet. Once the CH receives the data, the critical data packet is broadcast to all the member vehicles of the clusters. When the critical data packet is received at a member vehicle of the cluster, it drops the packet. On the other hand, the GW forwards the critical data packet to the next GW of the nearest cluster.

## 5. Simulation Environment and Results

The NS-3 and ndnSIM tools were used to implement the proposed scheme and analyze the performance. In addition to the traditional NDN structures, Simulation for Urban Mobility (SUMO) was employed to obtain the vehicle movements [31]. For a fair comparison, the simulations employed the same configuration, network configuration, Physical and MAC layer parameters, and mobility model. A total of 600 vehicles were considered moving at different speed rates. The length of the observed area was 300 m, and a three-lane road was considered. The remaining simulation settings are listed in Table 2.

In addition, a random number of vehicles (producers) were dispersed evenly over the network, and each vehicle randomly generated a critical data packet during the experiment. Every vehicle exchanged its location with all its neighbors through a pilot message.

Figure 6 shows the total number of critical data packets transmitted over each iteration with respect to the number of vehicles. We can observe that the naive scheme transmitted a massive number of critical data packets, while the proposed scheme transmitted a lower number of critical data packets. In the naive scheme, all the vehicles participate in broadcasting the critical data packets once it receives the data packets. As a result, every vehicle receives multiple copies of the same critical data packet. On the other hand, the proposed scheme allows only the CH to broadcast the critical data packet to all the vehicles in the cluster. Therefore, the proposed scheme transmits fewer data packets than the naive method.

In Figure 7, we can observe that the total number of critical data packets are generated and transmitted over time. It is noteworthy that the number of data packets by vehicles (producers) increases with respect to time. After starting the simulation, only one vehicle generates a critical data packet and transmits it. When the time reaches the 10th second, two vehicles generate critical data packets and transmit them, and the process continues. On the 100th second, about 98,000 and 18,000 critical data packets are observed to have been generated and transmitted by the naive and proposed schemes, respectively. The difference gets larger as the vehicles producing critical data packets increase. After 200 s, the total generated and transmitted data packets by the naive and proposed schemes rise to almost 350,000 and 6500, respectively. Therefore, the total number of transmitted critical data packets is multiplied by the number of vehicles that produce the critical data packets. The results show that the proposed scheme clearly transmitted fewer critical data packets than the naive method.

Figure 8 indicates the efficiency of the proposed and naive schemes. Efficiency is calculated by using Equation (Equation 10). We can observe that the efficiency of the proposed scheme is higher than the efficiency of the naive method. As the number of vehicles increases, the efficiency decreases. The straight line indicates that the number of vehicles remained almost the same due to the SUMO tool.
(10)$$Efficiency\left(\%\right)=\frac{RequiredDataPackets}{TotalTransmittedDataPackets}\times 100$$

In Figure 9, we observe that the number of critical data packets increases linearly with time. However, the number of vehicles is distributed randomly by applying the Poisson point distribution with a mean value of twenty (λ=20). The proposed scheme broadcasts a much lower number of critical data packets. On the other hand, the naive scheme broadcasts a massive number of critical data packets. The main reason is that only the CHs in the proposed scheme are responsible for broadcasting the critical data packets to every member vehicle within a cluster. However, in a naive method, every vehicle broadcasts the critical data packets and causes the broadcast storm during push-based data dissemination. Therefore, the proposed scheme mitigates the broadcast storm effect during push-based data forwarding within a VNDN environment.

Figure 10 demonstrates the total number of data packets broadcast with respect to the vehicles’ density. As the number of vehicles increases, the number of total data packets broadcast increases as well. The ratio between the proposed and naive schemes rises as the number of vehicles increases. In the naive method, the broadcast of critical data packets is directly proportional to the number of vehicles. In contrast, in the proposed scheme, the number of critical data packets is proportional to the number of clusters and CHs. As vehicles’ density increases, the number of clusters and CHs increases. As a result, the effect of the broadcast storm is mitigated because the CHs are responsible for broadcasting the critical data packets.

## 6. Conclusions

In VNDN, pull-based communication is basically supported. When a consumer requires content, it generates an interest packet (request) and broadcasts it to neighboring vehicles. The intermediate vehicles (routers) receive the interest, check the content in their CS, add the content to PIT along with its interface number, and broadcast it to their neighboring vehicles until it reaches the producer. The producer provides the content to the requested consumers using the same route. On the other hand, in push-based forwarding in VNDN, the producer generates critical content and broadcasts it to all the neighboring vehicles without any request; however, this creates a broadcast storm effect in the network. Therefore, we proposed the use of a fuzzy logic scheme to mitigate the broadcast storm effect in push-based data forwarding in VNDN. In our proposed scheme, fuzzy logic was designed to reduce the broadcast storm effect that could occur during the critical data packet broadcast in push-based data forwarding in VNDN. Critical circumstances were considered when a producer broadcast useful information to the rest of the vehicles. We employed the K-means clustering algorithm to divide the vehicles into groups. An Elbow algorithm was used to determine the optimum number of clusters. Then, the CH was selected using fuzzy logic. After the CH selection process, the producer would only forward the critical data packet to the CH. The CH would then exclusively broadcast it to all the member vehicles inside the cluster. In addition, the GW vehicles were responsible for the forwarding of the data packet to the GW vehicles of the neighboring clusters. As a result, the transmitted critical data packets were mitigated efficiently. The results show that the proposed scheme transmitted fewer data packets and reduced the chances of broadcast storms in the VNDN environment. It was also observed that as the number of vehicles increased, the total number of transmitted critical data packets also increased. For instance, if there are 50 vehicles, the total number of transmitted data packets is 1700 and 250 for the naive scheme and the proposed scheme, respectively. In terms of efficiency, the proposed scheme achieved more than 30% efficiency, while the naive method achieved less than 10% efficiency. In future work, we will consider both pull-based and push-based VNDN environments to obtain better results. In addition, efficient Radio Resource Management (RRM) in the VNDN environment is a hot topic to consider for future research.

## Figures and Tables

**Figure 1 sensors-22-08078-f001:**
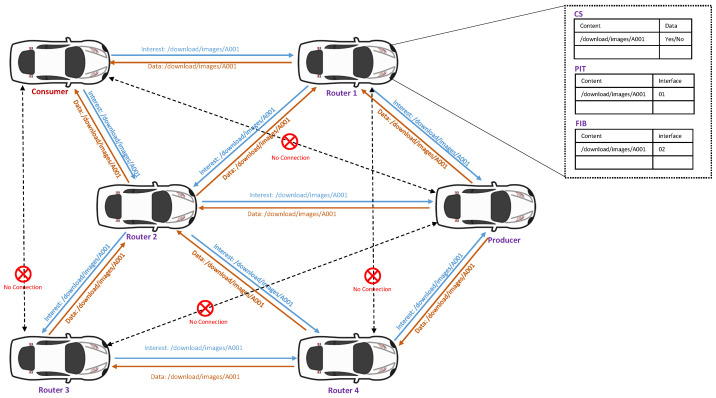
VNDN data forwarding architecture.

**Figure 2 sensors-22-08078-f002:**
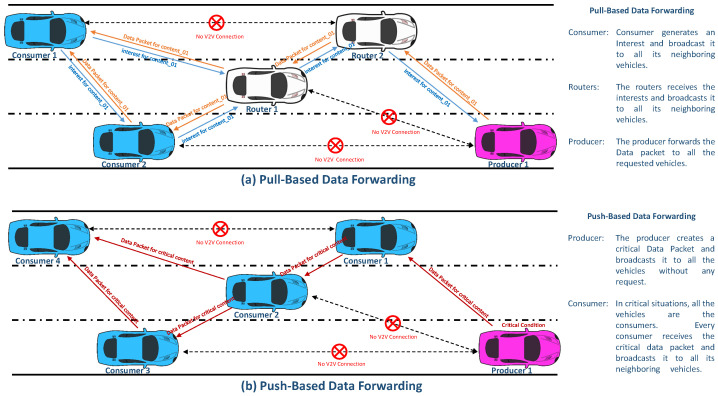
Comparison of pull-based and push-based data forwarding in VNDN.

**Figure 3 sensors-22-08078-f003:**
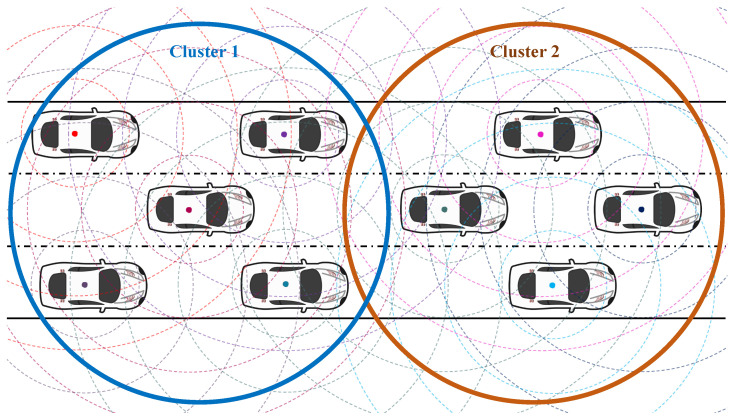
The proposed K-means clustering method for vehicle clustering in push-based data forwarding in VNDN.

**Figure 4 sensors-22-08078-f004:**
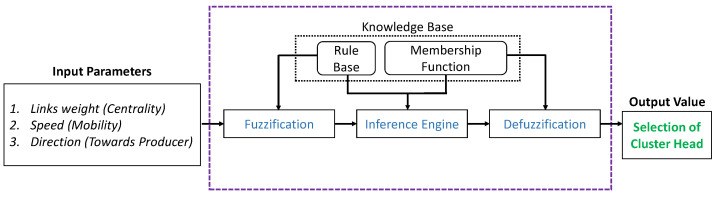
The fuzzy logic method used to select the CH.

**Figure 5 sensors-22-08078-f005:**
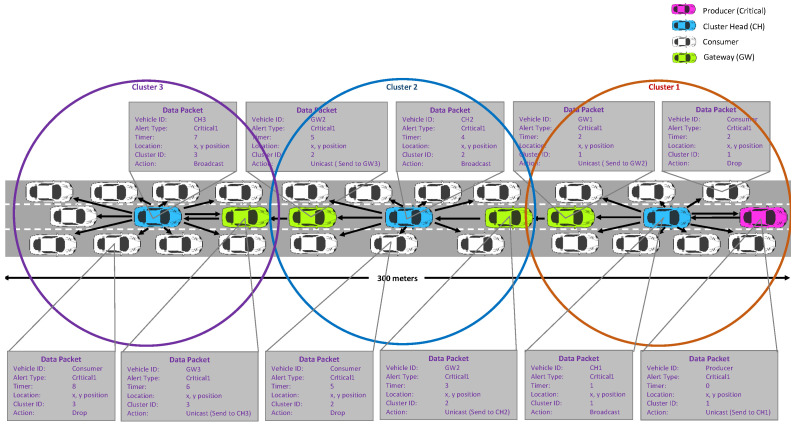
Example of a critical data forwarding scenario in the proposed push-based forwarding scheme using fuzzy logic method.

**Figure 6 sensors-22-08078-f006:**
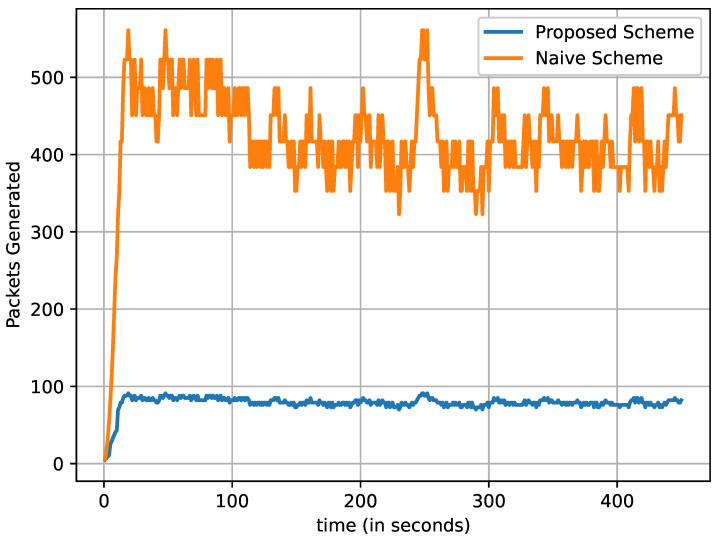
Total number of generated critical data packets at every iteration (time in seconds).

**Figure 7 sensors-22-08078-f007:**
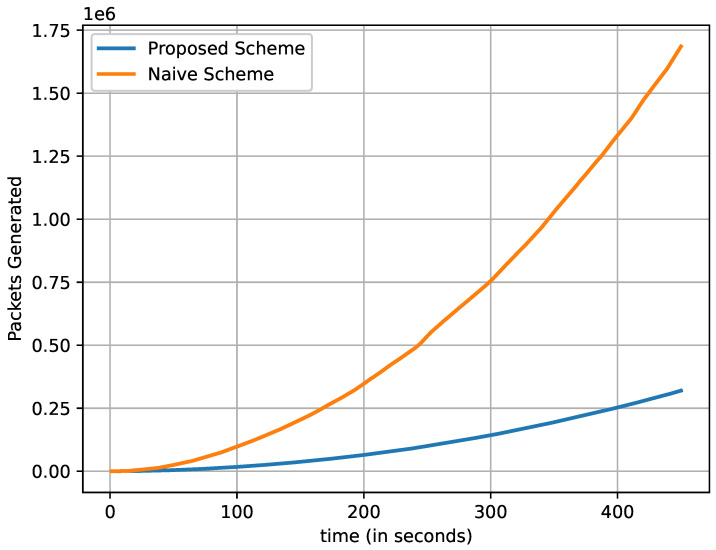
Total number of transmitted critical data packets with respect to the increasing number of vehicles (time in seconds).

**Figure 8 sensors-22-08078-f008:**
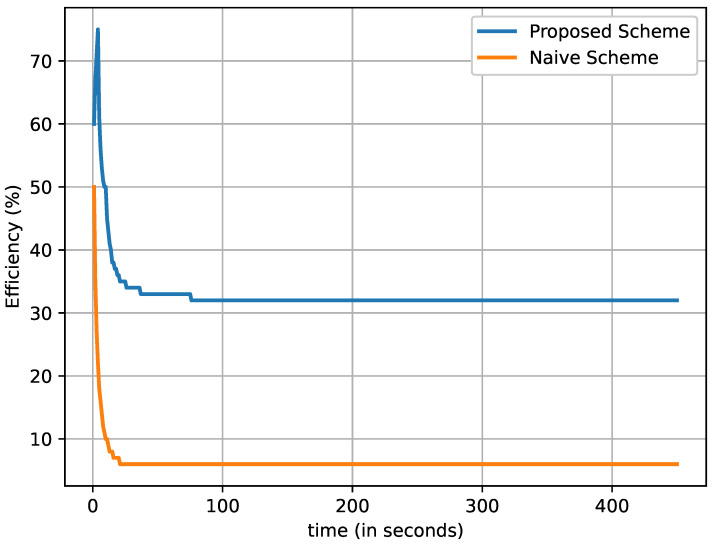
Efficiency in percentage.

**Figure 9 sensors-22-08078-f009:**
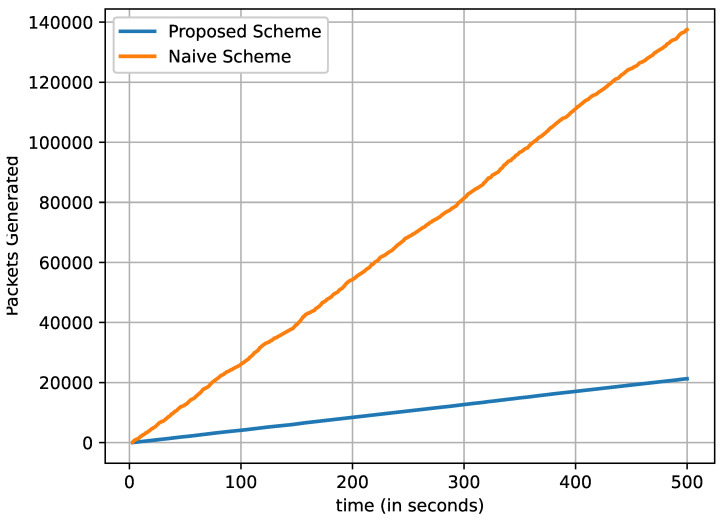
Total number of data packets broadcast with respect to the vehicle density; i.e., (λ=20).

**Figure 10 sensors-22-08078-f010:**
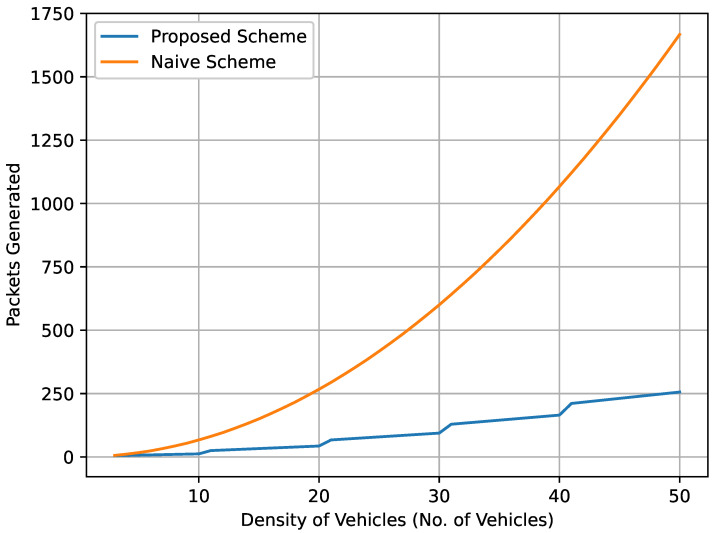
Total number of data packets broadcast with respect to the increasing number of vehicles.

**Table 1 sensors-22-08078-t001:** The proposed data packet format.

Field Name
Content
Data
Alert Type (Critical or Normal)
Location
Timer
Signature

**Table 2 sensors-22-08078-t002:** Simulation parameters and values.

Parameter	Value
Simulator	NS-3 and ndnSIM
Mobility Model	SUMO
Network Size	1–50 vehicles
Poisson Point Distribution (λ)	20
Covered Area	300 m
Transmission Range (m)	100
Vehicle Speed	30–180 KM/H
Transmission Mode	V2V (IEEE 802.11p)
Alpha (α)	0.50

## Data Availability

Not applicable.

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
