# Peer review of "Novel Fuzzy Logic Scheme for Push-Based Critical Data Broadcast Mitigation in VNDN"

_sensors, 2022, doi:10.3390/s22208078_

Round 1

Reviewer 1 Report

The paper proposes a Fuzzy logic-based scheme to mitigate the broadcast  storm effect in vehicular NDN by using cluster heads as unique responsible for broadcasting the critical data packets to the other vehicles in the cluster, whereas a gateway has the role of forwarding the critical data packets to the nearest clusters. The simulation results show that the proposed scheme outperforms the naive method in terms of transmitted data packets and efficiency. 

However I have some comments:

-       The paper is sometimes difficult to follow and should be revised throughout trying to better connect the various part or grouping differently parts related to the same concept. Starting from the introduction, for example, it is not easy to understand how the authors distinguish among connected vehicles, automated vehicles, vanets, and related applications.

-       The literature review appears poor and limited. For example, the use of cellular networks in VNDN or the impact of radio access techniques is not considered at all, but they all have a not negligible impact that should be at least discussed. See, e.g. Cellular aided vehicular named data networking, 2014 or Efficiency and Fairness Oriented Dynamic Task Offloading in Internet of Vehicles, 2022, Radio access for future 5G vehicular networks, 2017 or 

Caching in Vehicular Named Data Networking, 2020 or Vehicular named data networking framework, 2020 or 

Toward Named Data Networking: An Approach Based the Internet of Things Cloud With Edge Assistance 2022. Even if some of these works (just as examples) are not specifically focused on what addressed by the authors, however, they are important part of the literature on this topic and should be discussed.

-       Most of the intro on vanets in section 2.1 should be moved in the Introduction.

-       In urban environment, where the coverage distance could be reduced, the CH could be also the GW. Have the authors tried if this has an impact on the performance?

-       In the results, the authors consider a covered area of 300m and a tx range of 100m but this seems quite low, especially in highway or rural environment. The considered technology is 802.11p but which is the adopted MCS? What would change if 5G-V2X would be adopted?

-       Results have been presented in terms of number of generated packets and related efficiency of the algorithm, but the ratio of correctly received packets have never been considered or discussed. In this case, of course, also the channel and the scenario have an impact.

-       What about the latency of the proposed solution?

Reviewer 2 Report

This is an interesting article. One of the challenges of push-based data forwarding in VNDN is the broadcast effect, which occurs when each vehicle broadcasts critical information over the network. In the proposed scheme, the K-Means clustering algorithm is used to group vehicles into clusters, and then a fuzzy logic method is used to select cluster heads (CH).

1. How many cars does each cluster contain? Is there any basis?

2. Is this Producer customized by the system, or is it random? Can it be random?

3. In Figure 9, the number of packets generated by Naive Scheme is predicted or an actual experiment.

4. In Figure 4, the output value is Cluster Head, which is not clear, please explain, and please modify Figure 4 if necessary.

5. Please show the fuzzy rules of your experiment. Some rules should be displayed in the description of the article. If you are not sure, please refer to Fuzzy of MATLAB.

6. Figure 10 does not indicate the coordinate unit, please correct it.

Round 2

Reviewer 1 Report

The authors modified the paper accordingly to the reviewers' comments. The paper results improved and can be considered for publication in this journal.

Reviewer 2 Report

The author has tried his best to explain, and many questions have been supplemented.

Reviewer 3 Report

Thanks for Revision. No further comments.